# Enhancing Triage Efficiency and Accuracy in Emergency Rooms for Patients with Metastatic Prostate Cancer: A Retrospective Analysis of Artificial Intelligence-Assisted Triage Using ChatGPT 4.0

**DOI:** 10.3390/cancers15143717

**Published:** 2023-07-22

**Authors:** Georges Gebrael, Kamal Kant Sahu, Beverly Chigarira, Nishita Tripathi, Vinay Mathew Thomas, Nicolas Sayegh, Benjamin L. Maughan, Neeraj Agarwal, Umang Swami, Haoran Li

**Affiliations:** 1Department of Oncology, Huntsman Cancer Institute, University of Utah, Salt Lake City, UT 84112, USAumang.swami@hci.utah.edu (U.S.); 2Division of Medical Oncology, University of Kansas Cancer Center, Westwood, KS 66205, USA

**Keywords:** artificial intelligence, triage, emergency room, metastatic prostate cancer, ChatGPT 4.0, medical decision-making

## Abstract

**Simple Summary:**

Emergency rooms play a crucial role in providing immediate care to patients with metastatic prostate cancer. To enhance efficiency and accuracy of triage decisions for these patients, the authors conducted a retrospective analysis using ChatGPT 4.0, an advanced artificial intelligence system. The study investigated the effectiveness of ChatGPT in assisting healthcare providers with decision-making in the emergency room, focusing on patient outcomes and resource allocation. The findings demonstrated that ChatGPT showed high sensitivity in determining patient admission and provided accurate and comprehensive diagnoses. It also offered additional treatment recommendations, potentially improving the quality of care. These results suggest that ChatGPT has the potential to assist healthcare providers in enhancing patient triage and improving the efficiency and quality of care in emergency settings for patients with metastatic prostate cancer.

**Abstract:**

Background: Accurate and efficient triage is crucial for prioritizing care and managing resources in emergency rooms. This study investigates the effectiveness of ChatGPT, an advanced artificial intelligence system, in assisting health providers with decision-making for patients presenting with metastatic prostate cancer, focusing on the potential to improve both patient outcomes and resource allocation. Methods: Clinical data from patients with metastatic prostate cancer who presented to the emergency room between 1 May 2022 and 30 April 2023 were retrospectively collected. The primary outcome was the sensitivity and specificity of ChatGPT in determining whether a patient required admission or discharge. The secondary outcomes included the agreement between ChatGPT and emergency medicine physicians, the comprehensiveness of diagnoses, the accuracy of treatment plans proposed by both parties, and the length of medical decision making. Results: Of the 147 patients screened, 56 met the inclusion criteria. ChatGPT had a sensitivity of 95.7% in determining admission and a specificity of 18.2% in discharging patients. In 87.5% of cases, ChatGPT made the same primary diagnoses as physicians, with more accurate terminology use (42.9% vs. 21.4%, *p* = 0.02) and more comprehensive diagnostic lists (median number of diagnoses: 3 vs. 2, *p* < 0.001). Emergency Severity Index scores calculated by ChatGPT were not associated with admission (*p* = 0.12), hospital stay length (*p* = 0.91) or ICU admission (*p* = 0.54). Despite shorter mean word count (169 ± 66 vs. 272 ± 105, *p* < 0.001), ChatGPT was more likely to give additional treatment recommendations than physicians (94.3% vs. 73.5%, *p* < 0.001). Conclusions: Our hypothesis-generating data demonstrated that ChatGPT is associated with a high sensitivity in determining the admission of patients with metastatic prostate cancer in the emergency room. It also provides accurate and comprehensive diagnoses. These findings suggest that ChatGPT has the potential to assist health providers in improving patient triage in emergency settings, and may enhance both efficiency and quality of care provided by the physicians.

## 1. Introduction

Metastatic Prostate Cancer (mPC) is a significant global health issue and is the second leading cause of cancer-related deaths among men in the United States [1]. Patients with mPC often present to the emergency room (ER) with acute complications such as severe pain, spinal cord compression, pathological fractures, or urinary retention, necessitating prompt and accurate decision-making [2].

Effective triage in the ER is crucial to ensure patients receive appropriate and timely care. Rapid assessment and prioritization of patients based on the severity of their conditions can significantly impact outcomes, including morbidity and mortality rates [3]. However, ER triage, a complex and high-pressure task, can be challenging due to the diverse range of presenting conditions and swift decision-making requirements [4].

In recent years, Artificial Intelligence (AI) has emerged as a potent tool in healthcare, demonstrating the potential to enhance various aspects of patient care, including diagnosis, treatment planning, and patient triage [5]. AI algorithms can analyze vast amounts of data rapidly and accurately, providing valuable assistance to healthcare professionals in making critical decisions [6]. In the context of ER triage, AI can aid in accurately categorizing patients based on the severity of their condition, potentially reducing human error and improving efficiency [7].

The history of AI in medicine dates back to its early goals of providing decision-support tools. In the early years, the application of AI methods in medicine faced significant challenges due to limited availability of digital data. However, pioneering work by Weiss et al. in 1978 showcased the potential of AI in medical applications. They developed a glaucoma counseling program using the causal-associational network (CASNET) model, which demonstrated the feasibility of using AI to assist in the diagnosis and management of glaucoma [8]. As the field of AI progressed, researchers began exploring its application in analyzing oncological data. One of the notable milestones in this area was the development of the Cancer Distributed Learning Environment (CANDLE) initiative by the National Cancer Institute (NCI) in the early 2000s. The CANDLE initiative aimed to create a collaborative environment that enabled the sharing and analysis of cancer-related data, fostering the application of AI methods to enhance cancer research and patient care [9].

The use of AI in the analysis of prostate cancer patients has also gained significant attention. In recent years, machine learning algorithms have been employed to improve the diagnosis, risk stratification, and treatment planning for prostate cancer. For instance, deep learning algorithms have been employed to predict overall survival in patients with metastatic castration-resistant prostate cancer [10]. Additionally, radiomics models based on AI have been developed to predict biochemical recurrence in prostate cancer patients using follow-up MRI [11]. These applications highlight the potential of AI to provide valuable insights and support clinical decision-making in emergency settings for cancer patients [12].

One notable example of AI’s application in emergency medicine is the development of AI algorithms to aid in the triage of patients with cancer-related emergencies [13,14]. These algorithms leverage machine learning techniques to analyze patient data and assist healthcare providers in rapidly categorizing patients based on the severity of their condition [15]. By accurately identifying patients requiring immediate intervention, these algorithms can help prioritize care and ensure timely interventions.

In the context of machine learning, there are several relevant machine learning methods. One prominent approach is deep learning, a subset of machine learning that utilizes artificial neural networks to extract complex patterns and features from large datasets. Deep learning models, such as convolutional neural networks (CNNs), have demonstrated remarkable performance in medical image analysis, including the detection and classification of cancerous lesions [16]. Another valuable machine learning method is natural language processing (NLP), which focuses on the understanding and generation of human language. NLP techniques enable the extraction of meaningful information from clinical text, such as electronic health records and medical literature, to support decision-making [17]. In addition to these methods, recent research has also introduced innovative algorithms in the field of machine learning. For instance, Huang et al. proposed an algorithm of nonparametric quantile regression, providing a new perspective for handling complex medical data [18]. Lukauskas et al. developed a reduced clustering method based on inversion formula density estimation, offering an alternative approach for data analysis and pattern recognition [19]. Wang et al. presented a new algorithm for support vector regression with automatic selection of hyperparameters, improving the efficiency and accuracy of regression tasks [20].

ChatGPT 4.0, developed by OpenAI, is a state-of-the-art NLP model that has shown promise in various healthcare applications. Leveraging its ability to understand and generate human-like text, ChatGPT can assist healthcare providers in interpreting patient symptoms, formulating diagnoses, and deciding on suitable treatment plans [21]. Furthermore, during triage, it could potentially streamline the process, improve diagnostic accuracy, and optimize resource allocation. The current study investigates the effectiveness of ChatGPT 4.0 in assisting ER providers with decision-making for patients with mPC presenting to the ER. In addition, we focus on evaluating the potential of ChatGPT to improve patient outcomes and resource allocation by comparing its performance with standard ER practices.

## 2. Materials and Methods

### 2.1. Study Design and Patient Selection

This retrospective study utilized the AI model ChatGPT 4.0 to assess its effectiveness in triage decision-making for mPC patients presenting to the ER. Patients with a diagnosis of mPC who visited the University of Utah ER during the study period were included. Exclusion criteria were patients who had other types of cancer, those who visited the ER at another hospital, and cases with missing documentation regarding their ER visit.

### 2.2. Data Collection

Data collection occurred in two phases. In the first phase, a comprehensive chart review was conducted for patients who visited the ER between 1 May 2022 and 30 April 2023. This review included patient demographic information, medical history, and details of the ER visit. Structured data, such as baseline characteristics, including age, ethnicity, pathology types, tumor metastasis and co-existing conditions were collected.

In the second phase, data from electronic health records (EHR) were retrospectively collected. The EHR system used was Epic which contained various types of information documented by the ER physician, including chief complaints, history of present illness, laboratory results and radiology reports. ChatGPT 4.0 was provided with a comprehensive set of patient information collected during the first phase. However, the AI model was not provided with any information regarding the ER treatment course, consultation recommendations, or the ER physician’s analysis and decisions to ensure an unbiased evaluation.

ChatGPT 4.0 was then prompted to analyze this data and make decisions regarding patient triage. The AI model’s decisions regarding patient admission or discharge, Emergency Severity Index (ESI) score, and treatment recommendations were documented for further analysis.

This study was conducted following approval from the Institutional Review Board (IRB) at the University of Utah (IRB_00164099). All procedures were performed in compliance with relevant laws and institutional guidelines, and patient data were anonymized to protect privacy and confidentiality.

### 2.3. Outcomes

The primary outcome of this study was the sensitivity and specificity of ChatGPT 4.0 in determining whether a patient required admission or discharge. Sensitivity was defined as the proportion of true positive cases (correctly identified by ChatGPT) among patients who were eventually admitted. Specificity was defined as the proportion of true negative cases (correctly identified by ChatGPT) among patients who were eventually discharged from the ER.

Secondary outcomes included the concordance between ChatGPT 4.0 and ER physicians on admission diagnosis and treatment plan, the comprehensiveness of diagnoses, the length of Medical Decision Making (MDM) complexity, and the utility of the Emergency Severity Index (ESI) score generated by ChatGPT in predicting admission from the ER, hospital stay, and ICU stay.

### 2.4. Statistical Analysis

To evaluate the performance of ChatGPT in determining patient admission and discharge decisions, sensitivity and specificity were calculated. The agreement between the diagnoses and treatment plans provided by ChatGPT and ER physicians was evaluated using Cohen’s Kappa coefficient. The Wilcoxon signed-rank test was used to compare the number of diagnoses made by ER physicians and ChatGPT. Partial Correlation analysis was used to measure the degree of association between ESI generated by ChatGPT and hospital admission, length of hospital stay, and ICU admission. The length of MDM in the admission provided by ChatGPT and ER physicians was compared using a paired *t*-test.

In addition, a sensitivity analysis was conducted to explore whether changes in the number of diagnoses or the length of MDM complexity influenced ChatGPT’s admission predictions. Statistical analyses were conducted using the R software package. In all analyses, a *p*-value of less than 0.05 was considered statistically significant.

## 3. Results

### 3.1. Patient Characteristics

We initially screened 147 patients, of which 56 met the inclusion criteria. The screening process and study design are described in Figure 1, which illustrates the sequential flow of patients through different stages of our study. Following the initial screening, we excluded patients with other types of cancer, those who visited the ER at another hospital, and cases with missing documentation regarding their ER visit. A total of 91 patients were excluded based on these criteria. The analysis involved presenting the documented patient’s history of present illness, laboratory findings, and imaging reports to ChatGPT. However, we deliberately excluded the consultation note and the physician’s medical decision-making note from the information provided to ChatGPT. These notes were excluded to ensure that ChatGPT’s analysis remained separate from the medical decision-making process. It is worth noting that the same information used for ChatGPT analysis was also available to the ER physicians, serving as the “golden standard” for their decision-making regarding patient admission or discharge. Both ChatGPT and the ER physicians were therefore assumed to be acting upon the same set of information when making their respective medical decisions. 

The baseline characteristics of the patients included in the study were as described in Table 1. The majority of the patients were Caucasian (85.7%), with a median age of 75 years (ranging from 50 to 87 years). Most patients had adenocarcinoma (98.2%) with a median Gleason score of 8 (ranging from 6 to 10). The most common metastatic sites were the bone (69.6%) and lymph nodes (37.5%). In terms of Eastern Cooperative Oncology Group (ECOG) performance status, 19.6% had an ECOG score of 0, 28.6% had a score of 1, and 51.8% had a score greater than 1, indicating varying levels of functional impairment. Several coexisting conditions were observed among the patients, including diabetes (19.6%), hypertension (26.8%), hyperlipidemia (12.5%), depression (10.7%), atrial fibrillation (14.3%), chronic heart failure (7.1%), and gastroesophageal reflux disease (10.7%).

### 3.2. Sensitivity and Specificity of ChatGPT When Compared to ER Physicians

Out of the 23 patients admitted by the ER physicians, ChatGPT recommended admission for 22 patients (sensitivity: 95.7%) (Figure 2A). Out of the 33 patients discharged by the ER physicians, ChatGPT recommended discharge for six patients and recommended admission for 26 patients. There was one case where ChatGPT could not decide (specificity: 18.2%) (Figure 2B).

In the confusion matrix (Appendix A), ChatGPT made 22 true positive admissions, aligning with the decisions of ER physicians, and it made no false negative discharges. However, ChatGPT had 26 false positive admissions, admitting patients who could have been safely discharged.

### 3.3. Agreement and Comprehensiveness of Diagnoses between ChatGPT and ER Physicians

Some typical clinical scenarios assessed by ChatGPT and ER physicians are listed (Table 2). In some cases, there was an agreement between both parties, such as the diagnosis of a fall-related injury or urinary retention with complications associated with a Foley catheter. However, there were instances where discrepancies occurred, such as ChatGPT missing the suspicion of immunotherapy-related pancreatitis in a patient with abdominal pain. In most cases, ChatGPT showed more accurate diagnoses, such as a fall with soft tissue injury, an acute varus impacted right femoral neck fracture, ongoing gastrointestinal bleeding with worsening anemia, metastatic prostate cancer with suspected pain crisis, and accidental dislodgment of a percutaneous nephrostomy tube (Table 2).

The concordance on diagnoses between ChatGPT and ER physicians was assessed. In 87.5% of cases, ChatGPT provided the same primary diagnoses as the ER physicians (Cohen’s Kappa coefficient kappa = 0.33) (Figure 2C). Two physicians independently assessed the accuracy of terminology used by ChatGPT and ER physicians: Both parties achieved similar accuracy in 35.7% of cases, while ChatGPT had higher accuracy in 42.9% of diagnoses and ER physicians in 21.4% of diagnoses (*p* = 0.02 *) (Figure 2D). Furthermore, ChatGPT generated more comprehensive lists of diagnoses compared to ER physicians (median number of diagnoses: 3 vs. 2) (*p* < 0.001 ***) (Figure 3).

### 3.4. The Prediction Value of the ESI Score Generated by ChatGPT

ESI was a tool of triage commonly used in emergency departments to categorize and prioritize patients based on the severity of their condition and the required level of care. ESI scores calculated by ChatGPT were not associated with admission (*p* = 0.12), hospital stay length (*p* = 0.91), or ICU admission (*p* = 0.54) (Table 3).

### 3.5. Comparison of MDM Complexity

The MDM complexity between the ER physicians and ChatGPT was compared in two ways. Firstly, the number of words used in the assessment and plan was calculated (Figure 4). ChatGPT generated a shorter MDM length than the ER physicians (mean word count: 169 ± 66 vs. 272 ± 105, *p* < 0.001 ***). Secondly, we analyzed whether additional treatment was given in the MDM section. Despite shorter length in MDM, ChatGPT was more likely to provide additional treatment recommendations than ER physicians (94.3% vs. 73.5%, *p* < 0.001 ***).

### 3.6. Sensitivity Analysis

We conducted a sensitivity analysis to determine if changes in the number of diagnoses or the length of MDM complexity influenced ChatGPT’s admission predictions. When cases with fewer diagnoses (≤3) made by ChatGPT were analyzed, it had similar sensitivity (93.7%) and specificity (16.1%). When cases with shorter MDM (≤169 words) were analyzed, it also had similar sensitivity (100%) and specificity (20%).

## 4. Discussion

This study evaluated the performance of ChatGPT in the ER setting, exploring its potential as an aid in clinical decision-making. Our findings suggest that ChatGPT holds promise in certain areas, but improvements are necessary for reliable implementation in clinical practice.

A crucial aspect of ER practice is accurately determining the need for hospital admission. ChatGPT demonstrated high sensitivity in identifying patients requiring hospitalization, aligning closely with ER physicians’ decisions. However, the AI model’s specificity in discharging patients was significantly lower than that of ER physicians. These results indicate that while the model effectively recognized patients who needed admission, it struggled with accurately identifying those who could be safely discharged. Regarding diagnostic performance, ChatGPT matched the primary diagnoses made by ER physicians in 87.5% of cases. It also showed better diagnostic accuracy and produced more comprehensive diagnostic lists. This suggests that the AI model has the potential to serve as a valuable diagnostic tool in the ER setting, possibly providing a more holistic view of patients’ conditions.

Furthermore, ChatGPT was able to give additional treatment recommendations more frequently than ER physicians. This capability could be beneficial in ensuring comprehensive patient care, although it is essential to further explore the clinical relevance and appropriateness of these additional recommendations. In terms of ESI scores calculated by ChatGPT, there was no significant association between ESI and patient admission, the length of hospital stay, or ICU stay. This suggests that ChatGPT may not be as effective in predicting the severity of disease based on ESI scores. Finally, the use of ChatGPT resulted in a significantly shorter mean length of MDM compared to ER physicians. This could potentially enhance efficiency in ER settings, although the impact on patient outcomes needs to be further evaluated.

ChatGPT has the potential to revolutionize various aspects of healthcare, including ER triage and discharge processes. Patel et al. highlighted the significant promise of ChatGPT in generating discharge summaries, an area that often places a considerable burden on physicians [21]. Automating this process could reduce the workload and potentially improve the quality of discharge summaries, which have traditionally been prone to omissions [21]. Emerging research is also examining the use of ChatGPT in cancer care. Hopkins et al. studied the potential use of OpenAI’s ChatGPT as a virtual assistant for cancer patients, providing them with accessible information and answers [22]. ChatGPT’s capability to answer both simple and complex questions was compared to Google’s feature snippet, showing that while it can provide context-rich, conversational responses to queries about cancer, its answers can vary and may be incorrect, highlighting the importance of user caution and future improvements to ensure reliable information [22]. Uprety et al. revealed that ChatGPT’s capabilities include summarizing complex patient histories, making it particularly useful in oncology where patients undergo lengthy treatments, and interpreting Next-Generation Sequencing (NGS) reports to recommend personalized treatment plans, addressing the challenge of keeping up with rapidly evolving molecular oncology. Additionally, it can leverage ClinicalTrials.gov data to suggest relevant clinical trials for patients [23]. However, similar to Patel’s observation, implementing ChatGPT into clinical practice presents several challenges [21]. The AI model still relies on the information it receives, requiring salient data to be manually inputted. Other limitations include compliance with Health Insurance Portability and Accountability Act (HIPAA), inherent biases, and vulnerability to adversarial prompting pose significant challenges [23].

One of the most prominent issues with ChatGPT is hallucination [24]. Hallucination in AI, specifically in models like ChatGPT, refers to the generation of outputs that seem plausible but are factually incorrect or unrelated to the input context [25]. This is attributed to the model’s inherent biases, lack of real-world understanding, or training data limitations. In our study, all conversations were initiated independently to prevent interference from previous contexts. Despite this precaution, ChatGPT appeared confused about a case of UTI and began discussing the management of COPD instead. Hallucinations pose significant problems, including erosion of user trust, ethical issues, potential negative impact on decision-making, and possible legal implications [26]. Various mitigation strategies include improving training data, simulating adversarial scenarios, increasing the transparency and interpretation of the model, and incorporating human reviewers in the system.

In addition to ChatGPT, other chatbots are also being studied in healthcare. Lede et al. presented a chatbot named Tana [26]. It served as an administrative aid and a clinical helper during the height of the COVID-19 pandemic, answering frequently asked questions, facilitating appointment management, and gathering preliminary medical information prior to teleconsultations [27]. The use of Tana showcases the growing significance of chatbots in healthcare, particularly during periods of high demand or limited resources, such as a pandemic, offering a promising method to alleviate strain on healthcare systems and provide more efficient patient care. Dougall GPT is another AI chatbot tailored for healthcare professionals [28]. It provides clinicians with AI-tuned answers to their queries, augmented by links to relevant, up-to-date, authoritative resources. Besides facilitating the understanding and translation of medical literature, it assists in drafting patient instructions, consultation summaries, speeches, and professional correspondence, all while saving time compared to traditional search engines [27]. Wang et al. created Clinical Camel, an open-source healthcare-focused chatbot project that builds on the performance of fine-tuned large language model meta AI (LLaMa) with a combination of user-shared conversations and synthetic conversations derived from curated clinical articles [29]. It is notable that those Chatbots share the limitations of large language models (LLMs) in generating hallucinated outputs and potentially biased responses.

In addition to the limitations previously discussed, there are two significant limitations to consider regarding the use of chatbots like ChatGPT for actual medical purposes. Firstly, ChatGPT is not specifically designed or intended to be used for medical diagnosis, treatment, or decision-making. While it can provide general information and responses, it should not be relied upon as a substitute for professional medical advice or consultation with qualified healthcare providers. Secondly, the accuracy and reliability of AI-powered chatbots both heavily depend on the data they are trained on. If the developers’ data contains incorrect or outdated information, there is a risk that the chatbot may provide incorrect or misleading responses. It’s crucial to recognize that AI models, including chatbots, may not have real-time access to the most up-to-date medical knowledge or the ability to verify the accuracy of the information they provide. Given these limitations, it is essential to approach AI-powered chatbots in healthcare cautiously and use them as supplementary tools rather than definitive sources of medical information. Consulting healthcare professionals and authoritative medical sources remains crucial for accurate and reliable healthcare advice and decision-making.

## 5. Conclusions

In conclusion, our study demonstrated the potential of ChatGPT 4.0 as an AI language model for assisting emergency room (ER) physicians in the triage of patients with metastatic prostate cancer (mPC). By analyzing structured and unstructured data from electronic health records, ChatGPT provided valuable insights into patient characteristics and supported decision-making regarding patient admission or discharge. The AI model showed promising performance in accurately categorizing patients based on the severity of their condition, which can significantly impact patient outcomes and resource allocation in the ER setting. These findings highlight the ability of AI technologies to enhance efficiency and accuracy in healthcare decision-making processes.

## 6. Future Direction

To fully leverage the benefits of AI in healthcare, future research should focus on several important aspects. First, it is crucial to continue developing and validating accurate AI models that are specifically designed for medical purposes, ensuring their effectiveness, reliability, and compliance with regulatory guidelines such as HIPAA [30]. Additionally, the acceptance and trust in AI technology by both patients and healthcare providers need to be further explored and addressed. Efforts should be made to alleviate concerns about depersonalization and the potential loss of the human touch in healthcare interactions. Moreover, continuous efforts to mitigate the risks of technology failure and ensure the provision of correct and adequate information to patients are essential for the safe and effective implementation of AI in healthcare settings.

By advancing research in these areas, we can unlock the full potential of AI technologies in improving patient care, optimizing resource allocation, and ultimately enhancing healthcare outcomes. Further investigations and collaborations between AI experts, healthcare professionals, and regulatory bodies are needed to shape the future direction of AI integration in healthcare and ensure its successful and responsible application.

## Figures and Tables

**Figure 1 cancers-15-03717-f001:**
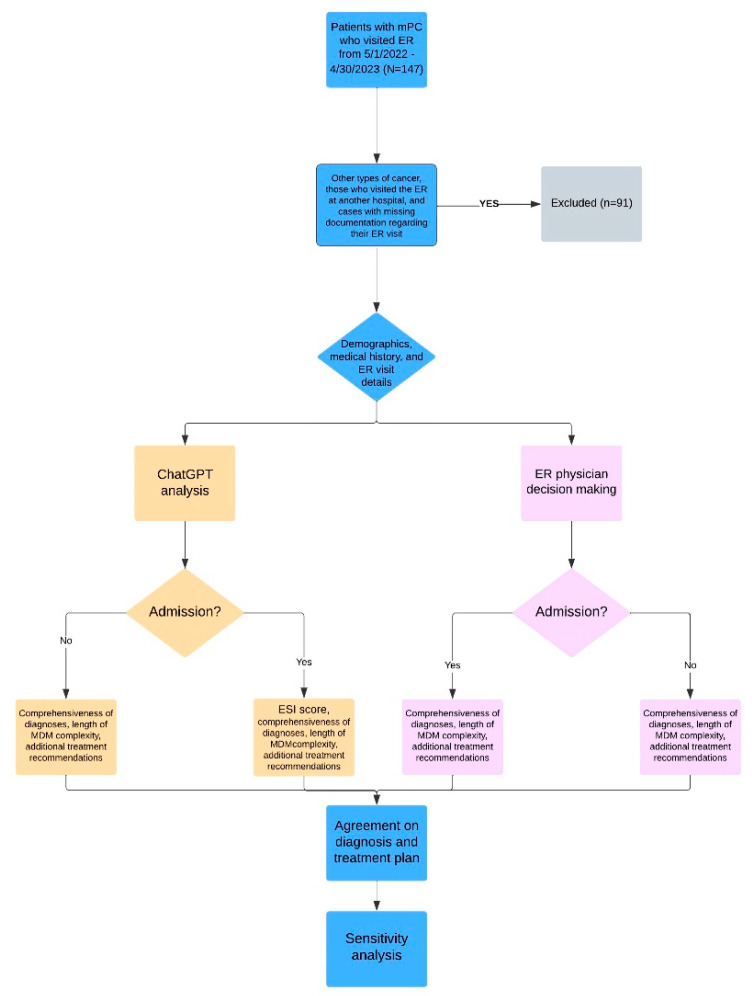
Flowchart of study design.

**Figure 2 cancers-15-03717-f002:**
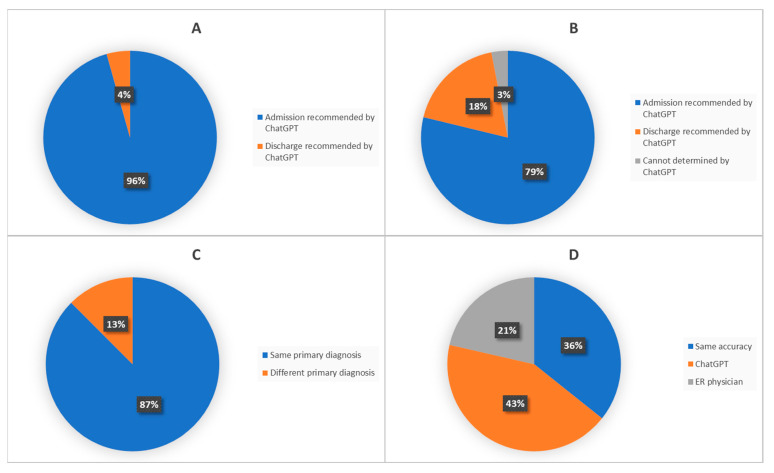
Comparing diagnostic decision made by ER physicians and ChatGPT. ((**A**): In patients admitted by ER physician; (**B**): In patients discharged by ER physician; (**C**): Agreement on diagnosis between ChatGPT and ER physician; (**D**): Who has more accurate diagnosis).

**Figure 3 cancers-15-03717-f003:**
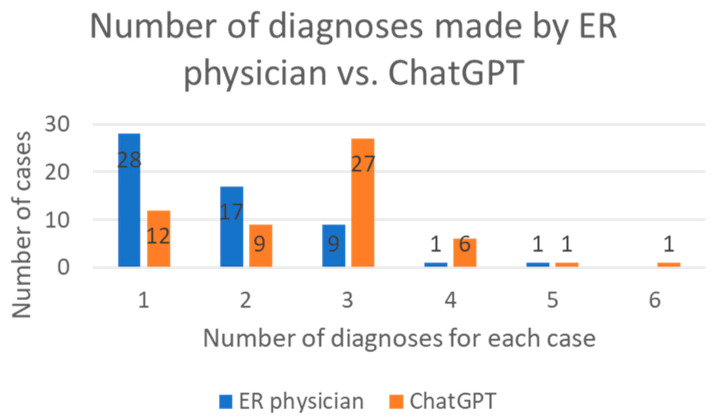
The number of diagnoses made by ER physicians and ChatGPT.

**Figure 4 cancers-15-03717-f004:**
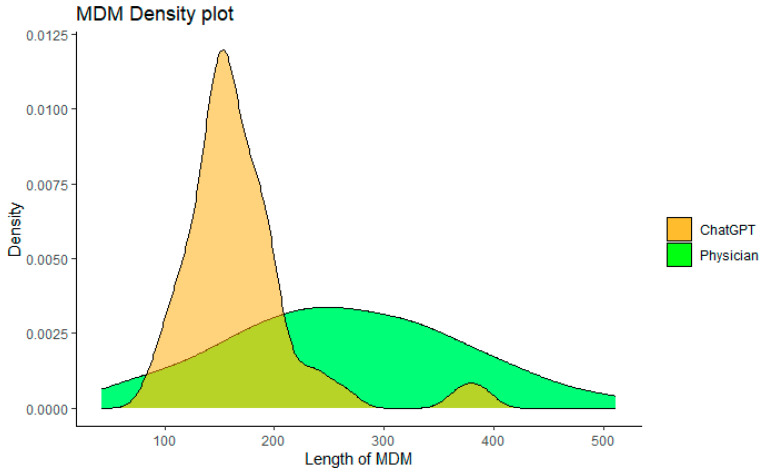
Comparing length of MDM written by ChatGPT and ER physicians.

**Table 1 cancers-15-03717-t001:** Patients’ characteristics at baseline.

Baseline Characteristics	
Age, median (range)	75 (50–87)
Race, *n* (%)	
Caucasian	48 (85.7)
African American	2 (3.6)
Hispanic	3 (5.4)
Asian	1 (1.8)
Hawaiian	1 (1.8)
Other	1 (1.8)
Adenocarcinoma, *n* (%)	55 (98.2)
Gleason score, median (range)	8 (6–10)
Metastatic sites, *n* (%)	
Bone	39 (69.6)
Lymph node	21 (37.5)
Lung	2 (3.6)
ECOG, *n* (%)	
0	11 (19.6)
1	16 (28.6)
>1	29 (51.8)
Co-exisiting conditions, *n* (%)	
Diabetes	11 (19.6)
Hypertension	15 (26.8)
Hyperlipidemia	7 (12.5)
Depression	6 (10.7)
Atrial fibrillation	8 (14.3)
Chronic heart failure	4 (7.1)
Gastroesophageal reflux disease	6 (10.7)

**Table 2 cancers-15-03717-t002:** Typical clinical scenarios and comparison of diagnoses made by ER physicians and ChatGPT.

Clinical Scenarios	Diagnoses Made by ER Physician	Diagnoses Made by ChatGPT	Comparison
67-year-old patient with a history of hypertension, obesity, overactive bladder, kidney stones, with mPC involving lymph nodes and bones, currently on a clinical trial with immunotherapy. He presented to ER for evaluation of abdominal pain.	Acute pancreatitis	Gastroenteritis	ER physician suspected possible immunotherapy-related pancreatitis, which ChatGPT did not notice. Pancreatitis was ruled out after admission.
87-year-old patient with a history of myocardial infarction, deep venous thrombosis (DVT) on rivaroxaban, acute kidney injury, and mPC presented with a CT scan demonstrating free air and mild duodenal thickening.	Peritonitis	Perforated peptic ulcer, peritonitis.	Final diagnosis after diagnostic laparoscopy: Pneumoperitoneum from perforated gastroduodenal ulcer.
62-year-old patient with a history of mPC, presented with shortness of breath.	Acute bronchitis due to rhinovirus	Acute respiratory distress, likely secondary to an asthma exacerbation, with a history of mPC.	Final diagnosis on discharge: asthma exacerbation and rhinovirus infection.
79-year-old patient with a history of amyloidosis heart failure, atrial fibrillation and mPC presented with nausea for one week.	Dehydration	Exacerbation of congestive heart failure	Patient had a history of heart failure, but the clinical picture was more consistent with dehydration from reduced oral intake
79-year-old patient with a history of DVT on wafarin, mPC presented with pain after a mechanical fall	Fall	Fall with soft tissue injury to the left arm and left knee	ChatGPT had a more accurate diagnosis.
72-year-old patient with a history of heart failure, stroke with residual left-sided deficits, DVT on warfarin, coronary artery disease (CAD), diabetes, hypertension and mPC status post radiation presented with multiple complaints.	Fatigue	Decompensated heart failure with hypotension.	Orthostatic hypotension was noted in the triage, which was not addressed in the ER physician’s assessment.
73-year-old patient with a history of a defibrillator in place for atrial fibrillation, ventricular tachycardia and mPC presented with chest discomfort.	Malaise	Unstable angina	Extensive cardiac workup was done after cardiology was consulted, though ChatGPT was not given the information about ER process.
67-year-old patient presented to the ER complaining of Foley catheter problems.	Obstruction of catheter	Urinary retention with complications associated with Foley catheter.	Both parties had the same diagnosis.
87-year-old patient with history of mPC, atrial fibrillation not currently on anticoagulation, heart failure with preserved ejection fraction, CAD, chronic kidney disease (CKD) who presented with concern for a fall.	Fall from ground level	Acute varus impacted right femoral neck fracture.	ChatGPT had a more accurate diagnosis.
75-year-old patient presented with sleepiness and weakness. Had a recent hospital stay for gastrointestinal (GI) bleeding.	Weakness	Suspected ongoing gastrointestinal bleeding with worsening anemia	ChatGPT had a more accurate diagnosis.
54-year-old patient with deafness, abdominal hernia, hepatitis C, and mPC to the spine and iliac crest presented for worsened pain.	Malignant neoplasm metastatic to bone	Metastatic prostate cancer with suspected pain crisis	Both parties had a similar diagnosis. ChatGPT had a more accurate diagnosis.
72-year-old patient with a history of chronic obstructive pulmonary disease (COPD), heart failure, CKD, hypertension, cirrhosis, stroke, mPC, chronic hydronephrosis, recurrent urinary tract infection (UTI), with bilateral percutaneous nephrostomy tubes, presented to the ER after his tube was displaced by accident.	Nephrostomy complication	Accidental dislodgment of the right percutaneous nephrostomy tube.	ChatGPT had a more accurate diagnosis.
69-year-old patient with heavily-treated mPC presented with weakness and inability to tolerate solid and liquid food	Failure to thrive, nausea and vomit	Dehydration	ChatGPT recommended hospice. Patient deceased after admission

**Table 3 cancers-15-03717-t003:** The relationship between the ESI score generated by ChatGPT and admission, hospital stay length and ICU admission.

	Hospital Admission	Hospital Stay Length	ICU Admission
Partial correlation coefficient	−1.59	−0.12	−0.62
*p* Value	0.12	0.91	0.54

## Data Availability

The data that support the findings of this study are available on request from the corresponding author US. The data are not publicly available due to them containing information that could compromise research participant privacy.

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
