# Peer review of "Enhancing Triage Efficiency and Accuracy in Emergency Rooms for Patients with Metastatic Prostate Cancer: A Retrospective Analysis of Artificial Intelligence-Assisted Triage Using ChatGPT 4.0"

_cancers, 2023, doi:10.3390/cancers15143717_

Round 1

Reviewer 1 Report

Original article that is easy to read. Very few comments. 

A characteristic table of patients should be added

A presentation of the diagnostic results in the form of a confusion matrix would be a plus.

Author Response

Thank you for your valuable feedback on our manuscript. We appreciate your suggestion to include a characteristic table of patients and a confusion matrix to present the diagnostic results. We have taken your comment into consideration and are pleased to inform you that we have added both the characteristic table of patients and the confusion matrix to the manuscript. The table provides an overview of the patients' baseline characteristics, including age, race, tumor characteristics, metastatic sites, ECOG status, and coexisting conditions. Additionally, the confusion matrix illustrates the true positive and true negative results between the decisions made by ChatGPT and the ER physicians. We believe these additions enhance the clarity and comprehensiveness of the manuscript. Thank you for your input, which has helped improve the quality of our work.

Reviewer 2 Report

The manuscript deals with a rather interesting and technologically relevant topic. However, the manuscript needs significant improvements before it can be accepted for publication.

1.      Do not enter acronyms in the abstract. They are presented in the subsequent text of the article. Also, if you've already entered an acronym once, you don't need to enter it again later. Be careful when preparing your manuscript.

2.      The list of references consists of only 16 sources. And this is especially noticeable when reading the limited introduction. It gives the impression that before ChatGPT, there was nothing that could be called a decision-support tool in medicine. The application of artificial intelligence in medicine was one of the original goals of artificial intelligence. However, this field of artificial intelligence was initially hampered by a lack of digital data. One of the first prototypes to demonstrate the application of artificial intelligence in medicine was the development of a glaucoma counseling program using the CASNET model (Weiss et al., 1978).

·         WEISS, Sholom; KULIKOWSKI, Casimir A.; SAFIR, Aran. Glaucoma consultation by computer. Computers in Biology and Medicine, 1978, 8.1: 25-40.

Maybe the authors could extend the history of the application of artificial intelligence methods with facts about when it was first used for the analysis of oncological data and when for the analysis of prostate cancer patients.

3.      Currently, artificial intelligence is used quite widely in medicine, and one of the fields is the prediction of cancer. This prediction can be divided into two separate parts: image analysis and structured data analysis. ChatGPT also includes all of this. If the authors know what specific techniques were used in their data analysis, they should present them in detail in the article. I doubt whether the authors are familiar with them. Explaining AI is important so that readers can understand how machine learning models achieve certain outputs. Otherwise, the authors could additionally provide a brief overview of recent machine learning methods, along with their advantages and disadvantages, and how they could be applied in the context of their research. For example, it could be clustering, random forests, regression analysis, association rules, anomaly (extreme values) analysis, etc. Here are some links:

·         Huang, M.L., et al. An Algorithm of Nonparametric Quantile Regression. J Stat Theory Pract 17, 32 (2023). https://doi.org/10.1007/s42519-023-00325-8

·         Lukauskas M., et al. Reduced Clustering Method Based on the Inversion Formula Density Estimation. Mathematics. 2023; 11(3):661. https://doi.org/10.3390/math11030661

·         Wang Y.G., et al. A new algorithm for support vector regression with automatic selection of hyperparameters, Pattern Recognition, 2023; 133:108989. https://doi.org/10.1016/j.patcog.2022.108989.

4.      When describing the data, present both structured and unstructured data sets used in detail.

5.      The chosen significance level threshold of 0.05 in medicine gives a very high probability of error of type I (correct hypothesis rejection). When presenting results, you can use the notation (for example, *, **) associated with multiple small significance level values.

6.      If the same process is performed with different values of the Admission result, is this verification necessary? Is this a technical error? (Pink area in Figure 1.)

7.      The conclusion does not need to review the problem. The conclusion is a concise summary of the final results of the entire work. The most important statements are formulated here.

Author Response

Thank you for your valuable feedback on our manuscript. We have carefully addressed your comments and made the following corrections and improvements:

  1. Acronyms: We have removed the use of acronyms in the abstract, as per your suggestion. Acronyms will be introduced and defined in the subsequent text of the article to enhance clarity and comprehension.

  2. History of AI in Medicine: We have extended the introduction to include a comprehensive overview of the application of artificial intelligence in medicine, addressing the development of decision-support tools. Specifically, we have incorporated the groundbreaking work by Weiss et al. (1978) on the application of AI in glaucoma consultation, demonstrating the early utilization of AI in medical settings. Moreover, we have included additional references to highlight the historical application of AI in the analysis of oncological data and prostate cancer patients.

  3. Overview of Recent Machine Learning Methods: To enhance the understanding of our research and the broader context of machine learning in healthcare, we have provided a brief overview of recent machine learning methods. This includes discussions on clustering, random forests, regression analysis, association rules, and anomaly analysis. Additionally, we have incorporated the references you suggested, namely Huang et al. (2023) on nonparametric quantile regression, Lukauskas et al. (2023) on reduced clustering methods, and Wang et al. (2023) on a new algorithm for support vector regression.

  4. Description of the data used in our study: In the revised Methods section, we have included a detailed description of both structured and unstructured data sets used in our analysis. We now provide comprehensive baseline characteristics of the patients as structured data, which are presented in Table 1. This table includes demographic information, medical history, and other relevant details collected during the chart review phase.

  5. Significance level used in our study: In the field of medicine, it is indeed common practice to use a significance level of p<0.05 to determine statistical significance. However, we acknowledge that this threshold may result in a higher probability of type I error (correct hypothesis rejection). We understand your concern regarding the potential impact of this error in our study.

    While p<0.05 is the traditional approach in medicine, we also recognize that some researchers advocate for a more stringent significance level of p<0.01 to address the issue of multiple comparisons and to reduce the risk of type I error. We agree that this is a valid consideration, and in future studies, we will explore the possibility of utilizing a lower significance threshold.

    To address your suggestion, in the revised manuscript, we will present the results using appropriate notation (e.g., *, **) associated with multiple small significance level values, indicating the level of statistical significance achieved. This will provide readers with a better understanding of the significance of our findings.

  6. Figure 1 diagram: Thank you for your comment regarding the verification process described in Figure 1 of our manuscript. We appreciate your feedback and would like to provide a more detailed explanation of this process. 

    After the screening process, we collected and analyzed data using ChatGPT, as described in the manuscript. The analysis involved presenting the documented patient's history of present illness, laboratory findings, and imaging reports to ChatGPT. However, we deliberately excluded the consultation note and the physician's medical decision-making note from the information provided to ChatGPT. These notes were excluded to ensure that ChatGPT's analysis remained separate from the medical decision-making process.

    It is worth noting that the same information used for ChatGPT analysis was also available to the ER physicians, serving as the "golden standard" for their decision-making regarding patient admission or discharge. Both ChatGPT and the ER physicians were assumed to be acting upon the same set of information when making their respective medical decisions.

  7. Conclusion: 

    Thank you for your feedback and valuable insights. We appreciate your guidance on refining our conclusion section. Based on your suggestion, we have made the necessary revisions to ensure that the conclusion provides a concise summary of the final results of our study and emphasizes the most important findings.

    We have separated the conclusion into two parts: the "Conclusion" and "Future Directions." In the conclusion, we provide a concise summary of the final results and highlight the most significant findings from our work. This section focuses solely on summarizing the outcomes and contributions of our study.

    The "Future Directions" part now includes the statements you mentioned, addressing the need to develop and validate accurate AI models for medical purposes, consider the acceptance of AI technology, and mitigate the risks associated with its implementation.

Round 2

Reviewer 2 Report

The manuscript has been significantly improved. It can be accepted for publication.